# The Key Role of Lung Ultrasound in the Diagnosis of a Mature Cystic Teratoma in a Child with Suspected Difficult to Treat Pneumonia: A Case Report

**DOI:** 10.3390/children9040555

**Published:** 2022-04-13

**Authors:** Elio Iovine, Laura Petrarca, Domenico Paolo La Regina, Luigi Matera, Enrica Mancino, Greta Di Mattia, Fabio Midulla, Raffaella Nenna

**Affiliations:** Department of Maternal Infantile and Urological Sciences, Sapienza University of Rome, 00161 Rome, Italy; elio.iovine@gmail.com (E.I.); laurapetrarca85@gmail.com (L.P.); laregina.domenico@gmail.com (D.P.L.R.); luigi.matera91@gmail.com (L.M.); enricamancino@gmail.com (E.M.); greta.di.mattia@gmail.com (G.D.M.); midulla@uniroma1.it (F.M.)

**Keywords:** mediastinal teratoma, lung ultrasound, mediastinal masses

## Abstract

To date, the diagnosis of mediastinal teratoma and mediastinal masses relies on the use of chest X-ray and CT. Lung and thoracic ultrasound is becoming increasingly used in the diagnosis and follow-up of many lung and thoracic diseases. Here, we report the case of a mature cystic teratoma in which the performance of lung ultrasound allowed to speed up the diagnostic workup and to provide the indication for the execution of CT of the thorax allowing the diagnosis.

## 1. Introduction

The anterior mediastinum represents an anatomic region site of several types of space-occupying masses, the most common of which are thymoma, teratoma, thyroid disease, and lymphoma. Of these, germ cell tumors and in particular teratoma, account for 15% of anterior mediastinal masses [1].

Teratomas are neoplasms composed of cellular elements derived from the three germinal layers, endoderm, mesoderm, and ectoderm. The localization appears to be age dependent; extragonadal tumors are more frequent in infants and early childhood, whereas ovarian tumors are more frequent in adolescents and adults [2]. The most frequent extragonadal site of this neoplasm is the sacrococcygeal area, followed by the anterior mediastinum [3].

In the past, the suspicion and diagnosis of a space-occupying mediastinal mass was exclusively based on chest X-ray and chest CT images. In fact, the mediastinum, delimited anteriorly by the sternum, posteriorly by the vertebrae, and surrounded by the ribs, was not considered explorable with ultrasound [4]. However, recently, ultrasound has been increasingly used in the diagnosis of lung and thoracic organ diseases, not only for infectious diseases [5,6].

We report a clinical case of a 13-year-old boy with an initial diagnosis of difficult to treat pneumonia in which the use of thoracic ultrasound allowed to make the hypothesis of mediastinal teratoma, later confirmed with further investigations.

## 2. Case Report

A 13-year-old boy was transferred to the Umberto I hospital in Rome from a peripheral hospital with a diagnosis of right medio-basal pneumonia complicated by pleural effusion. In his past medical history, he had a cerebral arteriovenous malformation that required multiple embolization procedures in previous years, the latest at the age of 10. On admission to the emergency department, the patient reported a fever lasting 3 days (highest temperature T 37.8 °C), right hemithorax pain and dyspnea. His vital sign reported a blood pressure of 108/70 mmHg, heart rate of 133 bpm, respiratory rate of 24 acts per minute, and room air oxygen saturation of 95%. His physical examination revealed pink skin and dry mucosae, forced orthostatic decubitus, normal cardiac activity, superficial breathing, and reduced air penetration in the right lung fields. There was good air penetration in the left chest. Superficial lymph nodes not increased in size.

First blood tests (Table 1) showed neutrophilic leukocytosis (WBC 14,800/μL) and a C-RP of 13.98 mg/dl. The patient underwent a chest X-ray that showed extensive and inhomogeneous hypodiaphania extending from the upper third to the lower third of the right lung compatible with pleural effusion and hypoexpanded upper lung pole. (Figure 1). Therefore, antibiotic therapy with Ceftriaxone was started.

In the hospital ward, blood cultures, procalcitonin levels, and nasal swabs for virus detection were performed and, due to the probable diagnosis of complicated pneumonia, Clindamycin was added to the therapy.

Blood cultures were negative for aerobic and anaerobic bacteria, procalcitonin was 0.37 ng/mL, and nasal swab tested positive for influenza B virus.

On the first day after the admission to the hospital ward, a lung ultrasound was performed, showing a pleural effusion at the right lung base of approximately 700 mL in volume. It also showed a thickening of parenchyma of most of the right lung with multiple abscess formations inside, partly confluent, with fluid-corpuscular content and rounded appearance. Absence of thickening or effusion on the left (Figure 2 and Figure 3).

During the second day of stay, the clinical conditions worsened with persistent fever, arterial saturation of 89–93% in air and dyspnea. Therefore, chest X-ray and lung ultrasound were repeated and oxygen therapy with nasocannulae 2 L/min was initiated.

Chest X-Ray showed persistence of large pleural effusion extending from the upper third to the lower third of the right hemithorax, unrecognizable right lung, verticalized course along the midline of the right main bronchus, and mediastinum slightly deviated to the left. Pulmonary ultrasound confirmed the presence of large pleural effusion and again showed at least two hypoechogenic and inhomogeneous oval-shaped formations measuring 43 × 30 mm and 12 × 12 mm, respectively (Figure 4, Figure 5 and Figure 6).

Due to the clinical and radiological deterioration, the radiological evidence of leftward mediastinal deviation, and the patient’s clinical unstable status, the patient was transferred to the intensive care unit for pleural drainage on the second day of stay.

Analysis of the pleural fluid revealed a cloudy appearance, increased LDH that was 952 mU/mL, amylase 320 U/L, and total protein 5.3 g/dL. Bacterioscopic and culture examinations resulted negative. According to Light criteria, because of the high LDH and proteins concentration in the pleural fluid, the effusion was therefore classified as an exudate.

In the following days, after the application of pleural drainage, due to the lack of clinical improvement despite antibiotic therapy, lung ultrasound was again repeated that once again shows most of the right lung as not aerated and numerous hypoechogenic oval formations with a fluid–fluid level inside with a corpuscular component at the base in the right hemithorax.

Six days after the admission to the hospital ward, in consideration of the peculiar ultrasound findings, which did not seem to suggest a simple pneumonia but rather an expansive or malformative lung or mediastinal disease, and in consideration of the lack of clinical improvement, a chest CT scan was performed.

The CT revealed a bulky, partially calcified, mass originating from the anterior mediastinum expanding into the right hemithorax. The mass caused a shift of mediastinal structures to the left and compression of the right lung parenchyma. The middle lobe bronchus was not visualized, likely because it was completely compressed by the mass (Figure 7, Figure 8 and Figure 9).

The diagnostic hypothesis based on the radiological findings was a cystic teratoma of the mediastinum or a cystic thymoma of the mediastinum.

The patient underwent surgery under general anesthesia for the removal of the mediastinal mass by sternotomy, and the tumor was successfully resected and sent for pathology. Histological examination was performed with hematoxylin and eosin staining, and diagnosis of a mediastinal germ cell tumor with the characteristics of a mature cystic teratoma was made.

## 3. Discussion

The mediastinum is an anatomical compartment located in the center of the thorax in which numerous space-occupying masses may occur or become manifest, both in the adult and pediatric populations. For a long time, ultrasound was not considered a suitable method for the analysis of this anatomical compartment and the masses originating in it because the bone structures surrounding it anteriorly and posteriorly prevent ultrasound penetration. In recent years, however, thoracic ultrasound, and in particular lung ultrasound, has emerged as a method for the study of the lung, not only in infectious disease but also for malformations, as in the case of congenital pulmonary airway malformation (CPAM), proving capable of identifying solid or cystic structures in the thorax, or more recently for the supportive diagnosis and follow-up of other chronic diseases affecting the lung such as asthma [7,8]. The presented case suggests a new possible application of lung ultrasound, which could be used as a supporting method also in the diagnosis of some mediastinal masses, such as teratoma.

Primary mediastinal teratoma is a relatively rare tumor, with cystic or solid appearance or a combination of the two and histologically composed of tissues derived from the three embryonic leaflets and foreign to the organ or anatomical site from which they originate [9].

Most teratomas, especially at pediatric age, are defined as mature teratomas and have a benign behavior. In contrast, immature teratomas, characterized by the presence of neuroectodermal-derived tissues, have an increased risk of malignancy and higher rates of local recurrence [10].

The clinical presentation of mediastinal disease can vary widely, with asymptomatic cases diagnosed incidentally and symptomatic cases accounting for approximately 60%. Symptoms are mainly due to the mass effect and compression of adjacent structures to the neoplasm and in particular of the airways leading to cough, wheezing, dyspnea, chest pain, cough, and stridor. Esophagus or superior vena cava may also be compressed with occurrence of dysphagia or superior vena cava syndrome [11].

In addition, the clinical scenario may be complicated by intracapsular hemorrhages, pleural or pericardial effusion with a case of difficult diagnostic interpretation that can easily be mistaken for infectious diseases [12,13].

In fact, the compression of bronchial or pulmonary structures and the resulting respiratory symptoms together with the compromised general condition and fever in case of major compression and possible complications could easily lead to a misdiagnosis of pneumonia that poorly responds to antibiotic therapy, as in the presented case.

A chest radiograph is usually the first examination performed in a patient with the reported symptoms. However, this examination is not sensitive for the diagnosis of teratoma, which may appear as a rounded or lobulated opacification extending from the anterior mediastinum beyond the midline that could be difficult to differentiate from pneumonia [14]. In addition, chest radiography does not provide precise indications about the anatomical relationships of the mass with other nearby structures and does not represent, especially in the pediatric population, a repeatable examination for follow-up of the disease.

Even if chest CT remains the key examination for the diagnosis of teratoma, allowing to describe its characteristics and the anatomical relationships with neighboring structures and to plan the surgical intervention, ultrasound could represent a suitable first step method for the diagnosis and characterization of mediastinal masses, especially in the pediatric population because of the lack of ionizing radiation. In fact, this examination allows to differentiate solid and cystic structures of the thorax and to describe the content of cysts [6,7,15,16]. In our case, for example, ultrasound described the presence of confluent cystic structures with corpuscular fluid content. In addition, lung ultrasound is much more sensitive and specific than chest X-ray in identifying pleural effusion often present in this kind of disease or in infectious lung diseases such as complicated pneumonias, especially in lying patients [17].

However, lung ultrasound has also some limitations such as not being able to investigate portions of the lung and mediastinum located under bony structures, not being able to properly visualize consolidations that do not reach the pleura surface, and being a very operator dependent method [18].

To date, there are no guidelines that include the use of lung ultrasound in the diagnostic workup of mediastinal masses and teratoma. Our case report is an example of how this method can be useful in the differential diagnosis of these diseases that can often mimic pneumonia and determine respiratory distress, allowing a prompt diagnosis. In our case, for example, the execution of lung ultrasound from the first day allowed us to observe a lack of improvement in the ultrasound picture over time despite appropriate antibiotic therapy and an atypical ultrasound image for pneumonia, permitting us to promptly suspect an alternative diagnosis and leading us to quickly perform CT of the chest.

The definitive therapy for mature teratoma is surgery, which allows to make a certain histological diagnosis and to solve complications and symptoms related to compression of thoracic structures with lifetime survival without recurrence close to 100% [19].

## 4. Conclusions

Thoracic and lung ultrasound may represent a useful tool for the differential diagnosis of pneumonia and respiratory distress in pediatrics.

This case demonstrates how lung ultrasound guided the diagnostic process and helped to place the indication for CT for the final diagnosis of mature cystic teratoma of the mediastinum with a pneumonia like presentation in a 13-year-old patient.

## Figures and Tables

**Figure 1 children-09-00555-f001:**
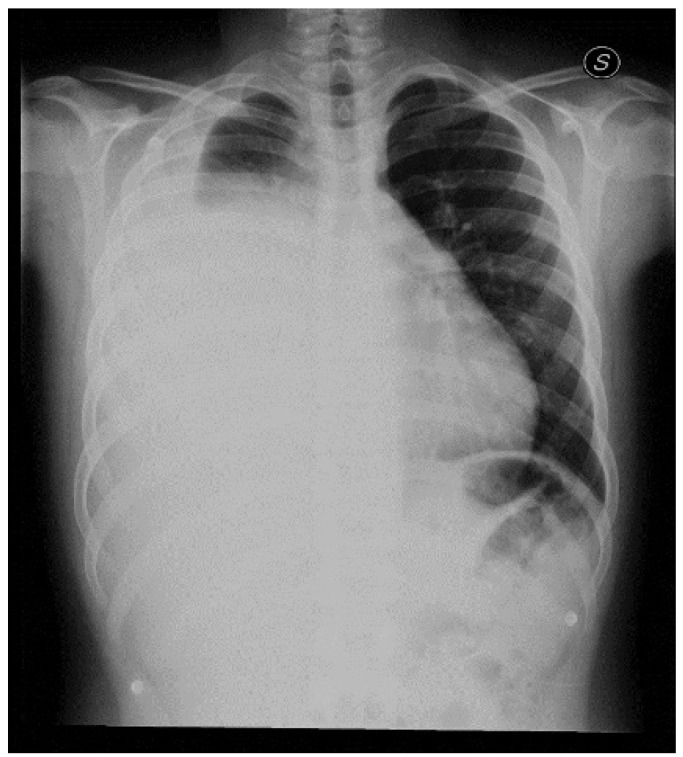
Chest X-ray performed in the emergency department. The exam showed an extensive and inhomogeneous hypodiaphania extending from the upper third to the lower third of the right lung as for pneumonia with a pleural effusion in the right hemithorax associated. The upper lung pole appeared hypoexpanded.

**Figure 2 children-09-00555-f002:**
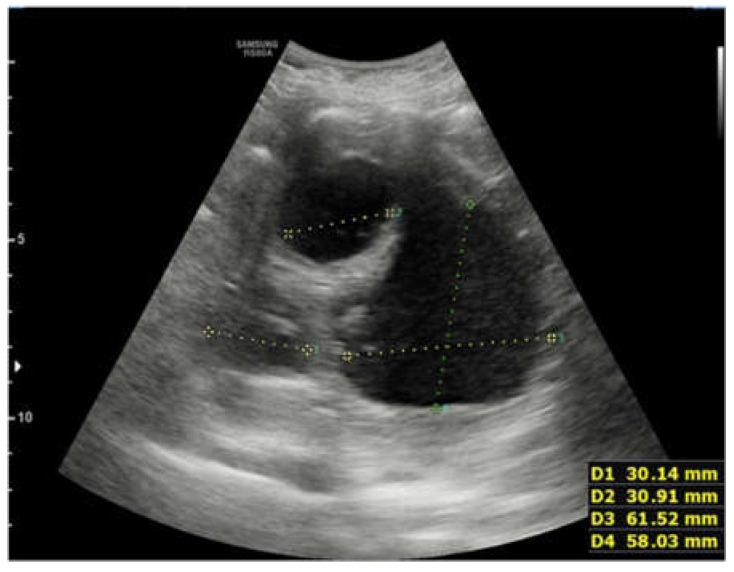
The first lung ultrasound performed on the first day after the admission to the hospital ward. The ultrasound showed a thickening of parenchyma of most of the right lung with multiple abscess formations inside, partly confluent, with fluid-corpuscular content and rounded appearance and an extensive pleural effusion at the right lung base.

**Figure 3 children-09-00555-f003:**
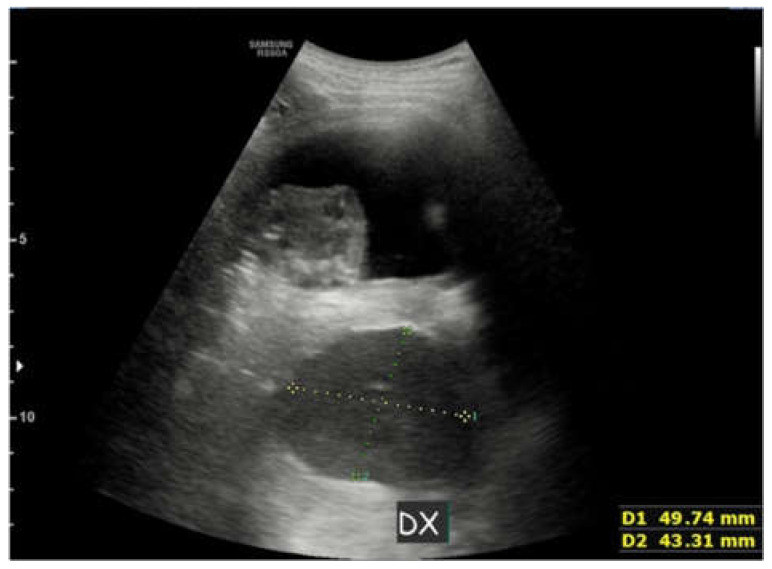
Image of the ultrasound performed on admission to the hospital ward. A pleural effusion and an oval formation of 49 × 43 mm is seen.

**Figure 4 children-09-00555-f004:**
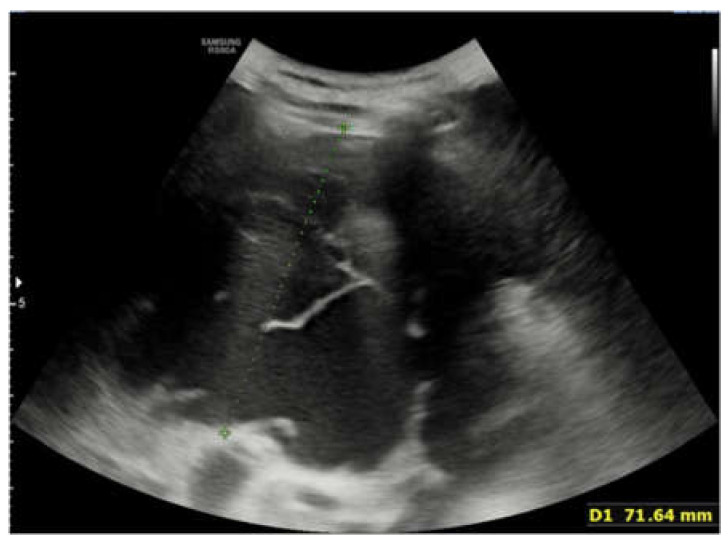
Lung ultrasound performed on the second day of stay, after the worsening of the clinical condition of the patient. The ultrasound showed the presence of large pleural effusion in the right hemithorax.

**Figure 5 children-09-00555-f005:**
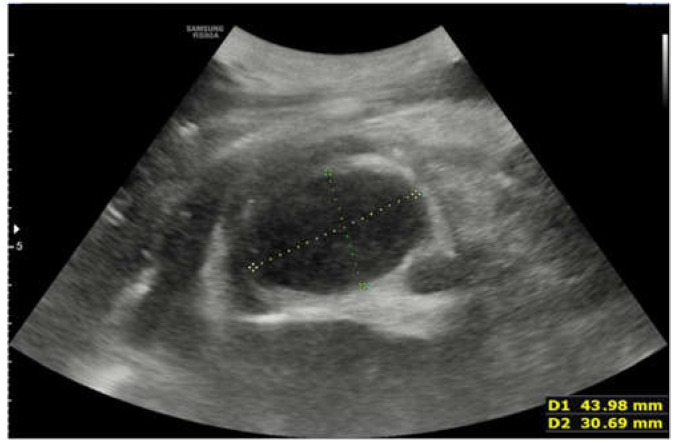
Ultrasound performed on the second day of stay. An hypoechogenic and inhomogeneous oval-shaped formations measuring 43 × 30 mm is seen.

**Figure 6 children-09-00555-f006:**
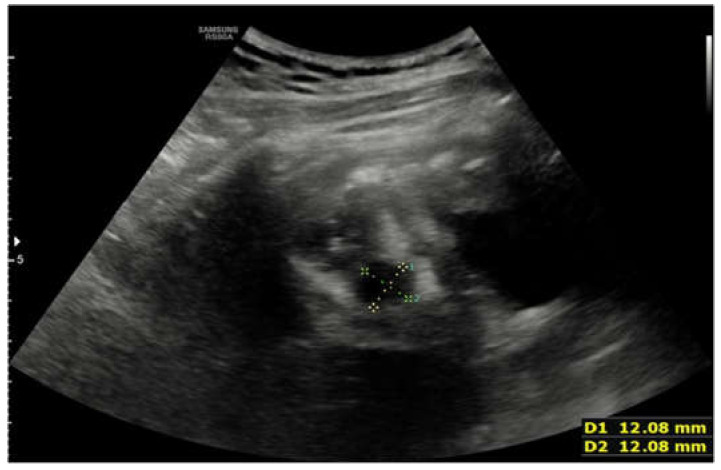
Ultrasound performed on the second day of stay. Another hypoechogenic and inhomogeneous oval-shaped formations measuring 12 × 12 mm is seen.

**Figure 7 children-09-00555-f007:**
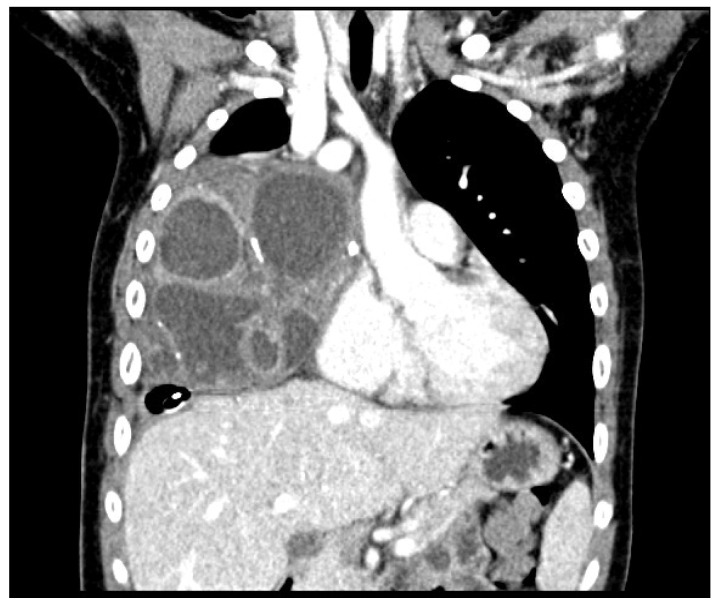
Frontal CT scan of the chest. A bulky and partially calcified mass originating from the anterior mediastinum and expanding into the right hemithorax is seen. The multi-chambered appearance of the mass appears similar to that observed on chest ultrasound. The mass caused a shift of the mediastinal structures to the left and compression of the right lung parenchyma. The middle lobe bronchus was not visualized, probably because it was completely compressed by the mass.

**Figure 8 children-09-00555-f008:**
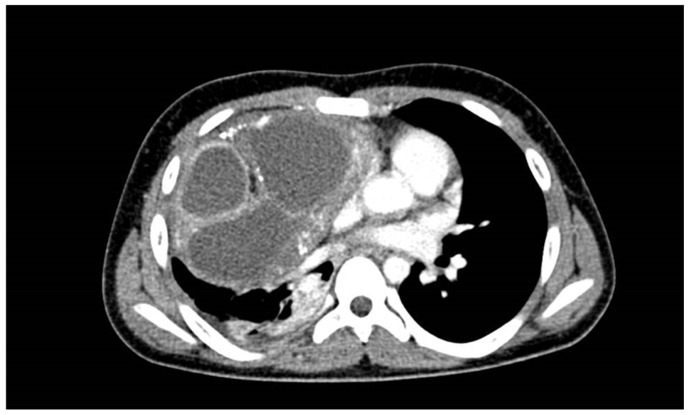
Transverse CT scan of the chest.

**Figure 9 children-09-00555-f009:**
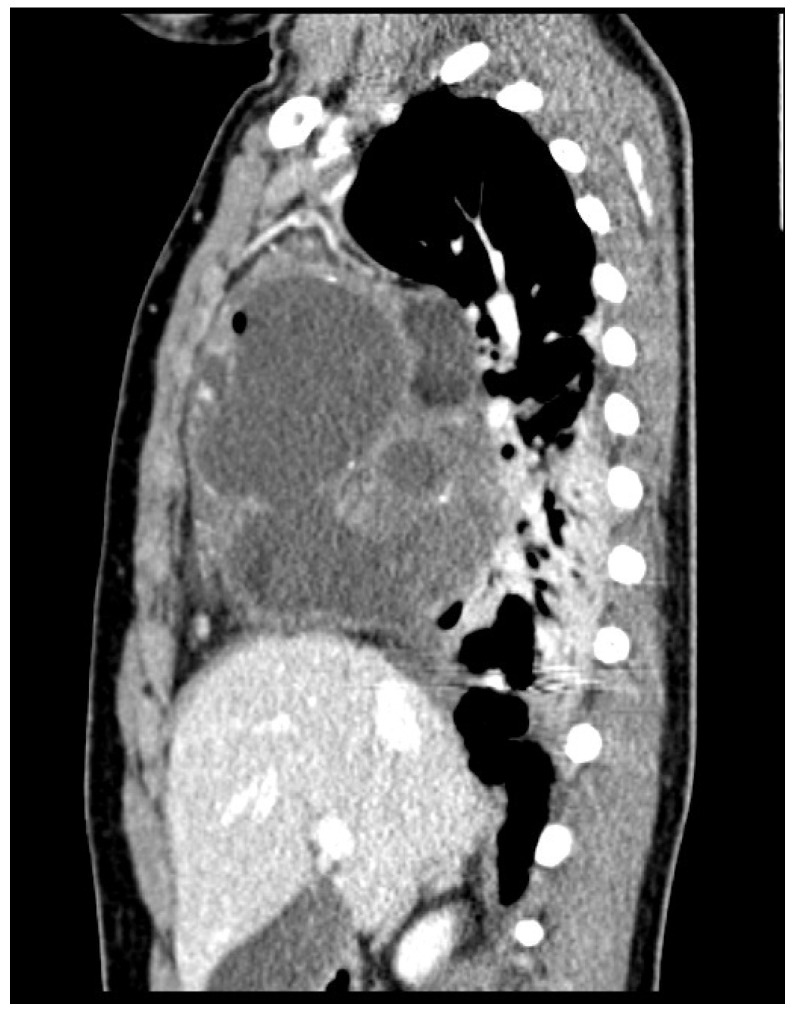
Sagittal CT scan of the chest.

**Table 1 children-09-00555-t001:** Blood test performed in the emergency department.

Test	Result	Reference Range	Measure Unit
Red Blood cells (RBC)	4.91	4.4–6	×10^12^/L
Hb	13.5	13–16	g/dL
Hct	39.5	36–51	%
MCV	80.5	80–100	fl
MCH	27.5	25–35	pg
MCHC	34.1	32–36	gr/dL
WBC	14.8	4.3–10.8	×10^9^/L
Neutrophils #	12.6	2–7	×10^9^/L
Lymphocytes #	1.2	1–4	×10^9^/L
Monocytes #	1	0–1.1	×10^9^/L
Eosinophils #	0.1	0–0.8	×10^9^/L
Basophils #	0.1	0–0.2	×10^9^/L
Neutrophils %	84.8	40–80	%
Lymphocytes %	7.9	20–40	%
Monocytes %	6.5	0–11	%
Eosinophils %	0.4	0–8	%
Basophils %	0.4	0–2	%
Platelet count	329	150–450	×10^9^/L
C-Reactive Protein	13.98	0–0.5	mg/dL

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
