# Peer review of "The Key Role of Lung Ultrasound in the Diagnosis of a Mature Cystic Teratoma in a Child with Suspected Difficult to Treat Pneumonia: A Case Report"

_children, 2022, doi:10.3390/children9040555_

Round 1

Reviewer 1 Report

I consider that this case is worth presenting, for its rarity and for disseminating the usefulness of thoracic ultrasound, the cheap, non-invasive, repeatable method.

Author Response

Reviewer 1:

I consider that this case is worth presenting, for its rarity and for disseminating the usefulness of thoracic ultrasound, the cheap, non-invasive, repeatable method

We thank the reviewer for his comment. We hope too that the use of this easy to use and inexpensive method will become more and more popular and that this article could also contribute. We have made some edits as also suggested by other reviewers to make the article clearer and with more images.

Reviewer 2 Report

Table 1 with lab test results – exclude

Figure 1 description – rewrite  The patient underwent chest X-ray that showed extensive and inhomo- 51 geneous hypodiaphania of the right mid-basal fields”…upper pole is also involved and pleural efussion is seen

Lack of second x ray image- it will be nice to see correlation with the ultrasound image

Lack of CT image as well

Case report section: too detailed..need to be more concise…

Discussion section

Need to write something about ultrasound limitations: e.g. subjective method – experience

Write also that ultrasound is more sensitive and specific in discovering pleural effusion then x ray, especially in lying patients

 Conclusion section: need to be more concise: tell that your case showed that ultrasound can be helpful method in pediatric population in making differential diagnosis of pneumonia…

References

Add this reference:

De Rose C, Miceli Sopo S, Valentini P, Morello R, Biasucci D, Buonsenso D. Potential Application of Lung Ultrasound in Children with Severe Uncontrolled Asthma: Preliminary Hypothesis Based on a Case Series. Medicines. 2022; 9(2):11. https://doi.org/10.3390/medicines9020011

Author Response

Reviewer 2:

Table 1 with lab test results – exclude

 We thank the reviewer. As suggested we modified the table keeping only the essential data and making it easier to read.

Figure 1 description – rewrite The patient underwent chest X-ray that showed extensive and inhomogeneous hypodiaphania of the right mid-basal fields”…upper pole is also involved and pleural effusion is seen

 As suggested by the reviewer, we modified the description of the x-ray highlighting the involvement of the upper lobe of the lung. (Page 2, line 52-53).

Lack of second x ray image- it will be nice to see correlation with the ultrasound image

Lack of CT image as well

 We thank the reviewer for this comment.  Unfortunately the image of the second x-ray is missing and at the moment we cannot find it. However, as suggested by the reviewer, we have added three CT images of the patient that, as rightly pointed out by the reviewer, appear essential for the proper interpretation of the clinical case and for the correlation with the ultrasound.

Case report section: too detailed..need to be more concise…

Thanks to the reviewer for the suggestion. We have now shortened the case report section trying to make it more concise.

Discussion section

Need to write something about ultrasound limitations: e.g. subjective method – experience

Write also that ultrasound is more sensitive and specific in discovering pleural effusion then x ray, especially in lying patients

As suggested by the reviewer, we briefly described some limitations of lung ultrasound and mentioned its superiority over chest X-ray in detecting pleural effusion. (Page 7, line 232-239)

 Conclusion section: need to be more concise: tell that your case showed that ultrasound can be helpful method in pediatric population in making differential diagnosis of pneumonia…

We agree with the reviewer. We have rewritten the conclusions to be more concise and focused on the concept of differential diagnosis with pneumonia. (Page 8, line 259-264)

References

Add this reference:

De Rose C, Miceli Sopo S, Valentini P, Morello R, Biasucci D, Buonsenso D. Potential Application of Lung Ultrasound in Children with Severe Uncontrolled Asthma: Preliminary Hypothesis Based on a Case Series. Medicines. 2022; 9(2):11. https://doi.org/10.3390/medicines9020011

Thanks to the reviewer for the suggestion. We added this new reference to the article. [8]

Reviewer 3 Report

The ultrasound images should contain captions that allow to appreciate the ultrasound alterations that the authors describe in the text. It is necessary to incorporate the tomographic image of the lesion so that the reader can make comparisons with the ultrasound findings.

The table on biological and biochemical values should be simplified and show only the most relevant values for the diagnostic orientation of the case.

Author Response

Reviewer 3:

The ultrasound images should contain captions that allow to appreciate the ultrasound alterations that the authors describe in the text. It is necessary to incorporate the tomographic image of the lesion so that the reader can make comparisons with the ultrasound findings.

We thank the reviewer for this comment. We agree with the reviewer that CT images are necessary for a good interpretation of the case and to correctly understand the role of ultrasound in this clinical case. Therefore, we added three CT images and improved the description in the captions of the ultrasound images.

The table on biological and biochemical values should be simplified and show only the most relevant values for the diagnostic orientation of the case.

We thank the reviewer for the comment. We have now simplified and condensed the table, keeping only the values useful for clinical case interpretation.

Round 2

Reviewer 3 Report

The changes made make the article more interesting.